# Hierarchical Bayesian modeling of multiregion brain cell count data

**Sydney Dimmock[1]\*, Benjamin MS Exley[2], Gerald Moore[3], Lucy Menage[4], Alessio Delogu[4], Simon R Schultz[3], E Clea Warburton[2], Conor J Houghton[1]\*, Cian O'Donnell[1,5]\***

[1]School of Engineering Mathematics and Technology, University of Bristol, Michael Ventris Building, Bristol, United Kingdom; [2]School of Physiology, Pharmacology and Neuroscience, University of Bristol, Biomedical Sciences Building, University Walk, Bristol, United Kingdom; [3]Centre for Neurotechnology and Department of Bioengineering, Imperial College London, South Kensington, London, United Kingdom; [4]Department of Basic and Clinical Neuroscience, Institute of Psychiatry, Psychology and Neuroscience, King's College London, London, United Kingdom; [5]School of Computing, Engineering and Intelligent Systems, Ulster University, Derry~Londonderry, United Kingdom

**\*For correspondence:**
sd14814.2014@my.bristol.ac.uk (SD);
conor.houghton@bristol.ac.uk (CJH);
c.odonnell2@ulster.ac.uk (CO'D)

**Competing interest:** The authors declare that no competing interests exist.

## eLife Assessment

This study proposes an **important** new approach to analyzing cell-count data, which are often undersampled and cannot be accurately assessed using traditional statistical methods. The case studies presented in the article provide **compelling** evidence of the superiority of the proposed methodology over existing approaches, which could promote the use of Bayesian statistics among neuroscientists. The authors have taken steps to make the methodology accessible, although some implementation difficulties are likely to remain.

**Abstract** We can now collect cell-count data across whole animal brains quantifying recent neuronal activity, gene expression, or anatomical connectivity. This is a powerful approach since it is a multiregion measurement, but because the imaging is done postmortem, each animal only provides one set of counts. Experiments are expensive, and since cells are counted by imaging and aligning a large number of brain sections, they are time-intensive. The resulting datasets tend to be undersampled with fewer animals than brain regions. As a consequence, these data are a challenge for traditional statistical approaches. We present a 'standard' partially pooled Bayesian model for multiregion cell-count data and apply it to two example datasets. These examples demonstrate that hierarchical Bayesian methods are well suited to these data. In both cases, the Bayesian model outperformed standard parallel *t*-tests. Overall, inference for cell-count data is substantially improved by the ability of the Bayesian approach to capture nested data and by its rigorous handling of uncertainty in undersampled data.

## Introduction

In studying the brain, we are often confronted with phenomena that involve specific subsets of neurons distributed across many brain regions. Computations, for example, are performed by neuronal networks connecting cells in different parts of the brain. As another example, from development,

**Figure 1.** Introduction. (**A**) Each of $N$ animals produces a cell count from a total of $R$ brain regions of interest. Cell-count data is typically undersampled with $N \ll R$. Scientists analyze the brain sections from the experiment for positive signals. Here, an example section is shown where teal points mark cells expressing the immediate early gene c-Fos (green and red lines indicate regions labeled as damaged). The final cell count is equal to the sum of these individual items sagittal brain map taken from the Allen mouse brain atlas: https://mouse.brain-map.org. (**B**) Partial pooling is a hierarchical structure that jointly models observations from some shared population distribution. It is a continuum that depends on the value of the population variance $\tau$. When $\tau = 0$, there is no variation in the population, and each individual observation is modeled as a conditionally independent estimate of some fixed population mean $\theta$ (complete pooling). As $\tau$ tends to infinity, observations do not combine inferential strength but inform an independent estimate $\gamma_i$ (no pooling). In between the two extremes, combine. Each observation can contribute to the population estimate while simultaneously supporting a local one to effectively model the variance in the data. The observed data quantities, $y_i$ to $y_n$, are highlighted with a thick line in the model diagrams. (**C**) An example of partial pooling on simulated count data. As the population standard deviation increases on the $x$-axis, the individual estimates $\exp(\gamma_i)$ trace a path from a completely pooled estimate to an unpooled estimate. Circular points give the raw data values. Parameters are exponentiated because the outcomes are Poisson and so parameters are fit on the log scale.

neurons in different anatomical regions of the brain share the same lineage. Data for each of these types of experiment will be considered here, but the challenge is very general: how to measure and analyze multiregion neuronal data with cellular resolution.

In a typical cell-count experiment, gene expression is used to tag the specific cells of interest with a targeted indicator (**Kawashima et al., 2014**). The brain is sliced, an entire stack of brain sections from a single animal is imaged, and the images are aligned and registered to a standardized brain atlas such as the Allen mouse atlas (**Lein et al., 2007**; **Oh et al., 2014**; **Daigle et al., 2018**; **Harris et al., 2019**), the images are segmented into anatomical regions, and the labeled cells in each region are counted. The resulting dataset consists of labeled cell counts across each of ~10–100 brain regions. This technology is being deployed to address questions in a broad range of neuroscience subfields, e.g.: memory (**Kim and Cho, 2017**; **Haubrich and Nader, 2023**; **Dorst et al., 2024**), neurodegenerative disorders (**Liebmann et al., 2016**), social behavior (**Kim et al., 2015**), and stress (**Bonapersona et al., 2022**).

Cell counts are often compared across groups of animals which differ by an experimental condition such as drug treatment, genotype, or behavioral manipulation. However, the expense and difficulty of the experiment mean that the number of animals in each group is often small. Ten is a typical number of samples for these experiments, but fewer is not uncommon. This means that these data are undersampled: the dimensionality of the data, which corresponds to the number of brain regions, is much larger than the number of samples, which usually corresponds to the number of animals (**Figure 1A**). Current statistical methods are not well suited to these nested wide-but-shallow datasets.

Furthermore, because of the complicated preparation and imaging procedure, there is often missing data along with variability derived from experimental artifacts.

In cell count data, there are two obvious sources of noise. The first of these is easy to describe: if a region has a rate that determines how likely a cell is to be marked for counting, then the actual number of marked cells is sampled from a Poisson distribution. The second source of noise is the animal-to-animal variability of the rate itself, and this depends on diverse features of the individual animal and the experiment that are often unrelated to the phenomenon of interest. The challenge is to control for outliers and 'poor' data points whose rate is noisy, while extracting as much information as possible about the underlying process. Dealing with outliers is often an opaque and ad hoc procedure. It is also a binary decision, a point is either excluded, so it does not contribute, or included, noise and all. This is where partial pooling helps. Partial pooling allows for the simultaneous estimation of

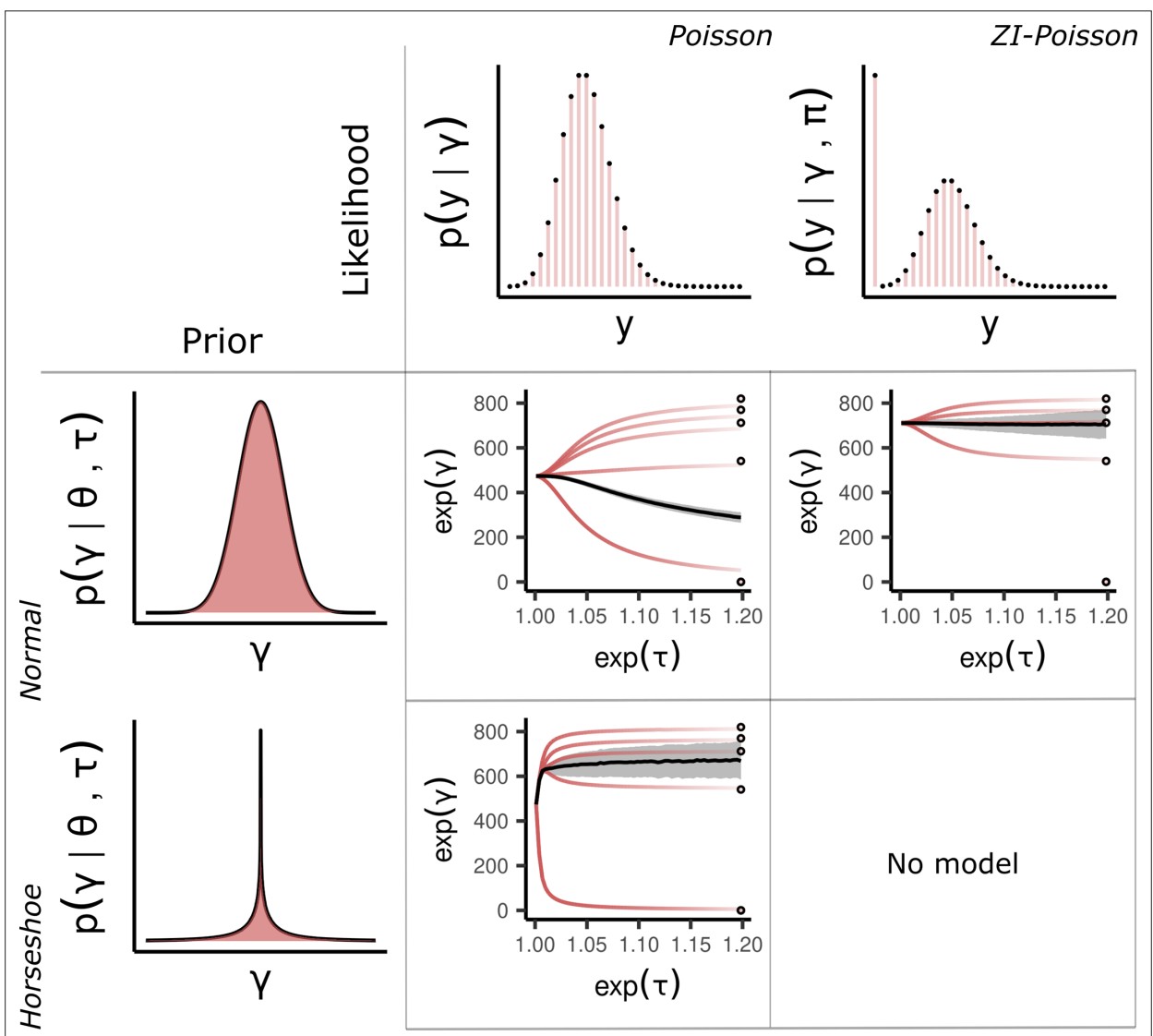

**Figure 2.** Methods. A table of partial pooling behavior for different likelihood and prior combinations. Rows are the two prior choices for the population distribution, and columns the two distributions for the data. Within each cell, the expectation of the marginal posterior $p(\exp(\gamma_i)|\theta, \tau, y)$ is plotted as a function of $\tau$. The solid black line is the expectation of the marginal posterior $p(\theta|\tau, y)$ with one standard deviation highlighted in gray. Top left: Combining a normal prior for the population with a Poisson likelihood is unsatisfactory in the presence of a zero observation. The zero observations influence the population mean in an extreme way owing to their high importance under the Poisson likelihood. Bottom left: By changing to a horseshoe prior, the problematic zero observations can escape the regularization machinery. However, regularization of the estimates with positive observations is much less impactful. Top right: A zero-inflated Poisson likelihood accounts for the zero observations with an added process, reducing the burden on the population estimate to compromise between extreme values. Bottom right: No model.

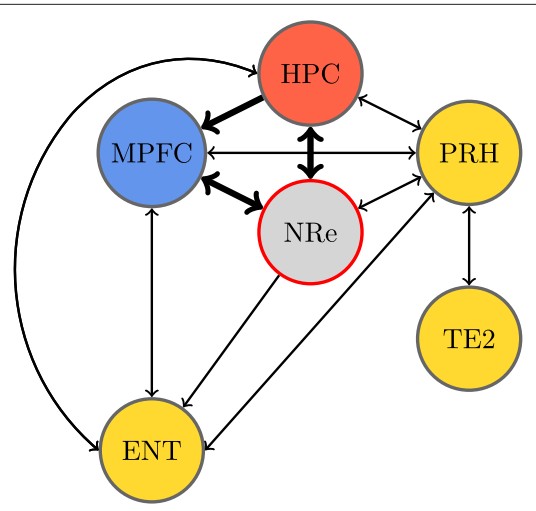

**Figure 3.** Recognition memory circuit. Schematic of the recognition memory network adapted from *Exley, 2019*. Bold arrows show the assumed two-way connection between the medial prefrontal cortex and the hippocampus facilitated by the nucleus reuniens (NRe). Colors highlight the hippocampus (HPC) (red), MPC (blue), and specific areas of the rhinal cortex (yellow). The NRe was lesioned in the experiment.

parameters describing individual data points and parameters describing populations. This helps the data to self-regularize and elegantly balances the contribution of informative and weak observations to parameter values (*Figure 2*).

In recent years, Bayesian approaches to data analysis have become powerful alternatives to classical frequentist approaches (*Gelman et al., 2013*; *McElreath, 2018*; *van de Schoot et al., 2021*). They have been applied to some types of neuroscience data, including neurolinguistics (*Dimmock et al., 2023*), neural coding (*Brown et al., 1998*; *Zhang et al., 1998*), synaptic parameters (*Moran et al., 2008*; *Costa et al., 2013*; *Bird et al., 2016*; *Bykowska et al., 2019*), and neuronal-circuit connectivity (*Mishchenko et al., 2011*; *Cinotti and Humphries, 2022*). A Bayesian approach is particularly well suited to cell-count data but has not previously been applied to this problem.

A Bayesian approach formalizes the process of scientific inference; it distinguishes the data and a probabilistic mathematical model of the data. This model has a likelihood which gives the probability of the observed data for a given set of model parameters. The model often has a hierarchical structure which we compose to reflect the structure of the experiment and the investigators' hypothesis of how the data depends on experimental condition. This hierarchy determines a set of a priori probabilities for the parameter values. The result of Bayesian inference is a probability distribution for these model parameters given the data, termed the posterior.

There are three advantages of a Bayesian approach that we want to emphasize: (1) while traditional multilevel models also allow a hierarchy (*Aarts et al., 2014*), Bayesian models are more flexible and the role of the model is clearer, (2) since the result of Bayesian inference is a probability distribution over model parameters, it indicates not just the fitted value of a parameter but the uncertainty of the parameter value. Finally, (3) Bayesian models tend to make more efficient use of data and therefore improve statistical power.

A Bayesian model also includes a set of probability distributions, referred to as the prior, which represent those beliefs it is reasonable to hold about the statistical model parameters before actually doing the experiment. The prior can be thought of as an advantage; it allows us to include in our analysis our understanding of the data based on previous experiments. The prior also makes explicit in a Bayesian model assumptions that are often implicit in other approaches. However, having to design priors is often considered a challenge, and here we hope to make this more straightforward by suggesting priors that are suitable for this class of data.

Here, our aim is to introduce a 'standard' Bayesian model for cell-count data. We illustrate the application of this model to two datasets, one related to neural activation and the other to developmental

lineage. For the second dataset, we also demonstrate a second example extension Bayesian model. In all cases, the Bayesian models produce clearer results than the classical frequentist approach.

# Materials and methods

## Data

To illustrate our approach, we consider two example applications, one which counts cells active in regions of the recognition memory circuit of rats during a familiarity discrimination task, and the other which examines the distribution of a specific interneuron type in the mouse thalamus.

### Case study 1 - Transient neural activity in the recognition memory circuit

The recognition memory network (*Figure 3*) is a distributed network which has been well studied using a variety of behavioral tasks. It includes the hippocampus (HPC) and perirhinal cortex (PRH), shown to deal with object spatial recognition and familiarity discrimination, respectively (*Barker and Warburton, 2011*; *Ennaceur et al., 1996*; *Norman and Eacott, 2004*); medial prefrontal cortex (MPFC), concerned with executive functions such as decision-making but also with working memory, and the temporal association cortex (TE2) used for acquisition and retrieval of long-term object-recognition memories (*Ho et al., 2011*). The nucleus reuniens (NRe) has reciprocal connectivity to both MPFC and HPC (*Hoover and Vertes, 2012*) and for this reason, it is also believed to be an important component of the circuit (*Barker and Warburton, 2018*; *Barker and Warburton, 2011*). In previous studies, lesions of the NRe have been shown to significantly impair long-term but not short-term object-in-place recognition memory (*Barker and Warburton, 2018*).

The data analyzed in this case study were collected to investigate the role of the NRe in the recognition memory circuit through contrasting the neural activation for animals with a lesion in the NRe with neural activation for animals with a sham surgery. The immediate early gene c-Fos is rapidly expressed following strong neural activation and is useful as a marker of transient neural activity. Animals in the experiment performed a familiarity discrimination task (single-item recognition memory), discriminating novel or familiar objects with or without an NRe lesion, and the number of cells that expressed c-Fos was counted in regions across the recognition memory circuit. The two-by-two experimental design allocated animals to each of the four experiment groups $\{\text{sham}, \text{lesion}\} \times \{\text{novel}, \text{familiar}\}$, and cell counts were recorded from a total of 23 brain regions. The visual cortex V2C and the motor cortex M2C were taken as control regions as they were not expected to show differential c-Fos expression in response to novel or familiar objects (*Exley, 2019*).

### Case study 2 - Ontogeny of inhibitory neurons in mouse thalamus and hypothalamus

The second dataset comes from a study (*Jager et al., 2021*) that, in part, counted the number of inhibitory interneurons in the thalamocortical regions of mouse. Sox14 is a gene associated with inhibitory neurons in subcortical areas. It is required for the development and migration of local inhibitory interneurons in the dorsal lateral geniculate nucleus (LGd) of the thalamus (*Jager et al., 2021*; *Jager et al., 2016*). Consequently, Sox14 is useful for identifying discrete neuronal populations in the thalamus and hypothalamus (*Golding et al., 2014*; *Jager et al., 2016*).

The experiment compared heterozygous (HET) and knockout (KO) mouse lines. The HET knock-in mouse line $Sox14^{GFP/+}$ marked Sox14-expressing neurons with green fluorescent protein (GFP); the homozygous KO mouse line $Sox14^{GFP/GFP}$, in contrast, was engineered to block the expression of the endogenous Sox14 coding sequence (*Delogu et al., 2012*). Each animal produced two samples, one for each hemisphere. In total, there are ten data points, six belonging to HET (three animals) and four to KO (two animals). Each observation is 50-dimensional, corresponding to 50 individual brain regions in each hemisphere.

## Hierarchical modeling

Our goal in both cases is to quantify group differences in the data. We present a 'standard' hierarchical model. This model reflects the experimental features common to cell count experiments and reflects the hierarchical structure of cell-count data; the standard model is designed to deal robustly and efficiently with noise. On some occasions, to reflect a specific hypothesis, the structure of a particular

experiment or an observed source of noise, this model can be further refined or changed to target the analysis. We will give an example of this for our second dataset.

At the bottom of the model are the data themselves, the cell counts $y_i$. The index $i$ runs over the full set of samples, which in this case comprises 23 brain regions × animals × groups ≈920 datapoints in the first study, and 50 brain regions × HET animals + brain regions × KO animals ≈ 500 datapoints in the second. The basic assumption the model makes is that this count is derived from an underlying propensity, $\lambda_i > 0$, which depends on brain region and, potentially, group:

$$y_i \sim Poisson(\lambda_i) \tag{1}$$

Hence, the propensity $\lambda_i$ is the mean of the Poisson distribution, and a statistical model is used to describe the dependence of this parameter on brain region and animal. Since $\lambda_i$ is strictly positive, a log-link function is introduced:

$$\log \lambda_i = \theta_{r[i],g[i]} + \gamma_i + E_i \tag{2}$$

where we have used 'array notation' (*Gelman and Hill, 2006*), mapping the sample index $i$ to properties of the sample, so $r[i]$ returns the region index of observation $i$, and similar for $g[i]$ but for groups and animals. The sample-by-sample variability is given by $\gamma_i$; this is modeled as Gaussian noise:

$$\gamma_i \sim Normal(0, \tau_{r[i],g[i]}) \tag{3}$$

whose size depends on region and group. This equation demonstrates a potentially surprising aspect of partially pooled models: the over-parameterization.

Ignoring $E_i$ for now, the rate has been split between two terms: $\theta_{r[i],g[i]}$ is the fixed effect, which is constant across animals, and $\gamma_{r[i],a[i]}$ is the random effect, which captures the animal to animal variability. While $\theta_{rg}$ models the mean for log cell count for each region, given the condition; $\gamma_i$ models variation around this mean. For this reason, $\gamma_i$ is assumed to follow a normal distribution with zero mean. The regression term may appear over-parameterized, without $\theta_{rg}$ the $\gamma_i$ could 'do the work' of matching the data. However, the model is regularized by a prior; observations with a weak likelihood will have their random effect $\gamma_i$ shrunk toward the population location. The amount of regularization depends on the variation in the population, a quantity that is estimated from each likelihood. This is how partial pooling works as an adaptive prior for 'similar' parameters (*Figure 1B*). The data 'pools' some evidence while still allowing for individual differences in samples.

The final term is the exposure $E_i$. Cell counts may be recorded from sections with different areas. The exposure term scales the parameters in the linear model as the recording area increases (*McElreath, 2018*). In our model, the exposure is equal to the logarithm of the recording area; this value is available as part of the experimental data.

The set of parameters $\tau_{r,g}$ models the population standard deviations of the noise for each region $r$ and animal group $g$. When working on the log scale, priors for these parameters are typically derived in terms of multiplicative increases. Since the parameters are positive, they are assigned a half-normal distribution

**Table 1.** Parameter table for the hierarchical model.

| Parameter | Description |
| --- | --- |
| $E_i$ | Exposure |
| $\kappa_i$ | Horseshoe inflation. |
| $\pi$ | Zero inflation |
| $\gamma_i$ | Random effect for observation $i$ |
| $\theta_{rg}$ | Fixed effect for region $r$ in group $g$ |
| $\tau_{rg}$ | Scale of random effects for region $r$ in group $g$ |

$$\tau_{r,g} \sim HalfNormal(\log(s)) \tag{4}$$

with an appropriately chosen scale $s > 1$. For our analyses, we used $s = 1.05$ because this gives a Half-Normal distribution with 95% of its density in the interval $[0, \log(1.1)]$. This translates into an approximate 10% variation around $\exp(\theta)$ at the upper end, which is a moderately informative prior, reflecting our belief that within-group animal variability is small relative to between-group variability. This regularization also helps model inference when the datasets are undersampled. *Table 1* gives a reference for all the model parameters.

## Horseshoe prior

Cell-count data often has outliers, for example, due to experimental artifacts. Since by default, the likelihood does not account for these outliers, they may cause substantial changes in fitted parameter values. This is demonstrated in *Figure 2*, where a careless application of the Poisson distribution on data with several zero counts has a large influence on the posterior distribution. There are two general options for dealing with outliers: either modeling them in the likelihood or in the prior. Although the likelihood option is preferred as it is more direct - see our zero inflation model below - it can be hard to design because it requires knowledge of the outlier generation process. The alternative is via a flexible prior such as the horseshoe (*Carvalho et al., 2010*; *Piironen and Vehtari, 2017*). This more generic option may be suitable as a default 'standard' approach in the typical case where outliers are poorly understood.

The horseshoe prior is a hierarchical prior for sparsity. It introduces an auxiliary parameter $\kappa_i$ that multiplies the population scale $\tau$. This construction allows surprising observations far from the bulk of the population density to escape regularization.

$$\gamma_i \sim Normal(0, \tau_{r[i],g[i]} \times \kappa_i) \tag{5}$$

$$\kappa_i \sim HalfNormal(1). \tag{6}$$

An example of this is given in *Figure 2* as the bottom left cell of the $2 \times 2$ table of models. The horseshoe prior often uses a Cauchy distribution, but in our case, the heavy tail causes problems for the sampling algorithm (see Appendix 1: Horseshoe densities).

## Zero inflation

A particular trait of the second dataset is that there are a large number of zero data points ($\sim 6\%$). Although a zero observation is always possible for a Poisson distribution, for plausible values of the propensity, zeros should be rare. It is likely that for some regions, the experiment has not worked as expected, and the zeros show that something has 'gone wrong' and that the readings are not well described by a Poisson distribution. Here, we extend the model to include this possibility. This is a useful elaboration of the standard model. In the standard model, the horseshoe prior ensures that these anomalous readings only have a small effect on the result, but it is more informative to extend the model to include them. While this particular extension is specific to these data, it also serves as an example of how a standard Bayesian model can serve as a starting point for an iterative investigation of the data.

The zero-inflated Poisson model is intended to model a situation where there are zeros unrelated to the Poisson distribution. In this case, this might, for example, be the result of an error in the automated registration process that identifies regions and counts their cells. It is a mixture model if

$$y_i \sim ZIPoisson(\pi, \lambda_i) \tag{7}$$

There is a probability $\pi$ that $y_i = 0$ and a $1 - \pi$ probability that $y_i$ follows a Poisson distribution with rate $\lambda_i$. Importantly, this means there are two ways in which $y_i$ can be zero, through the Bernoulli process parameterized by $\pi$ or through the Poisson distribution. This has the effect of 'inflating' the probability mass at zero with the additional parameter $\pi$ giving the proportion of extra zeros in the data that could not be explained by the standard Poisson distribution. This distribution can be visualized in *Figure 2*, and further mathematical details are described in Appendix 1: Distributions.

## Model inference

Posterior inference was performed with the probabilistic programming language Stan (*Carpenter et al., 2017*), using its custom implementation of the No-U-Turn (NUTS) sampler (*Betancourt, 2016*; *Hoffman and Gelman, 2014*). For each model, the posterior was sampled using four chains for 8000 iterations, with half of these being attributed to the warm-up phase. This gives a total of 16,000 samples from the posterior distribution.

## Results

We describe differences in estimated counts between groups in terms of $\log_2$-fold changes. Fold changes are useful because they prevent differences that are small in absolute magnitude from being masked by regions with high overall expression. Our results compare Bayesian highest density

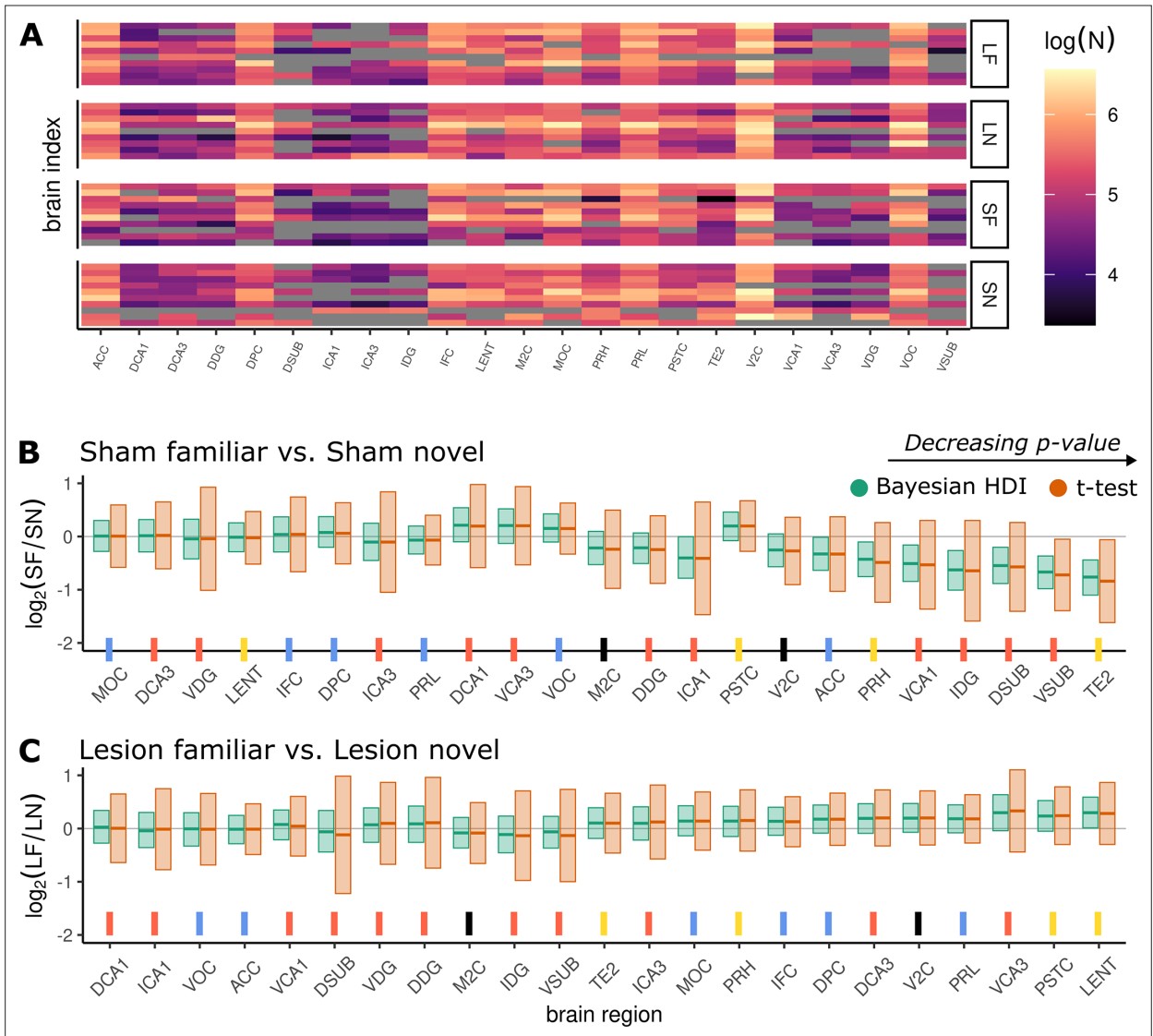

**Figure 4.** Results - Case study 1. (**A**) Heatmap of the raw log cell count data. Each row corresponds to a single animal, columns correspond to brain regions. Animals are grouped into lesion-familiar (LF), lesion-novel (LN), sham-familiar (SF), and sham-novel (SN). (**B, C**) $\log_2$-fold differences for each surgery type: B shows differences between SF and SN groups; C shows differences between LF and LN groups. The 95% Bayesian highest density interval (HDI) is given in green, and the 95% confidence interval calculated from a Welch's $t$-test in orange. Horizontal lines within the intervals mark the posterior mean of the Bayesian results, and the raw data means in the $t$-test case. The $x$-axis is ordered in terms of decreasing p-value from the significance test and ticks have been color-paired with the nodes in the recognition memory circuit diagram (*Figure 3*). Black ticks are not present in the circuit because they are the control regions in the experiment.

intervals (HDIs) with the confidence interval (CI) from an uncorrected Welch's *t*-test. The Bayesian HDI is calculated from the posterior distribution and is the smallest width interval that includes a chosen probability, here 0.95 (to correspond to $\alpha = 0.05$), and summarizes the meaningful uncertainty over a parameter of interest.

## Case study 1 - Transient neural activity in the recognition memory circuit

Results for the first dataset are presented in *Figure 4*. *Figure 4A* plots cell-count differences between the novel and familiar conditions without lesion and *Figure 4B* with lesion. These data were collected to investigate the role of different hippocampal and adjacent cortical regions in memory. However, some regions of interest, such as the intermediate dentate gyrus (IDG) and the dorsal subiculum (DSUB), look underpowered: for both regions, there is a markedly nonzero difference in expression between the novel and familiar conditions in the sham animals, but a wide CI overlapping zero makes the evidence unreliable (orange bars, *Figure 4A*).

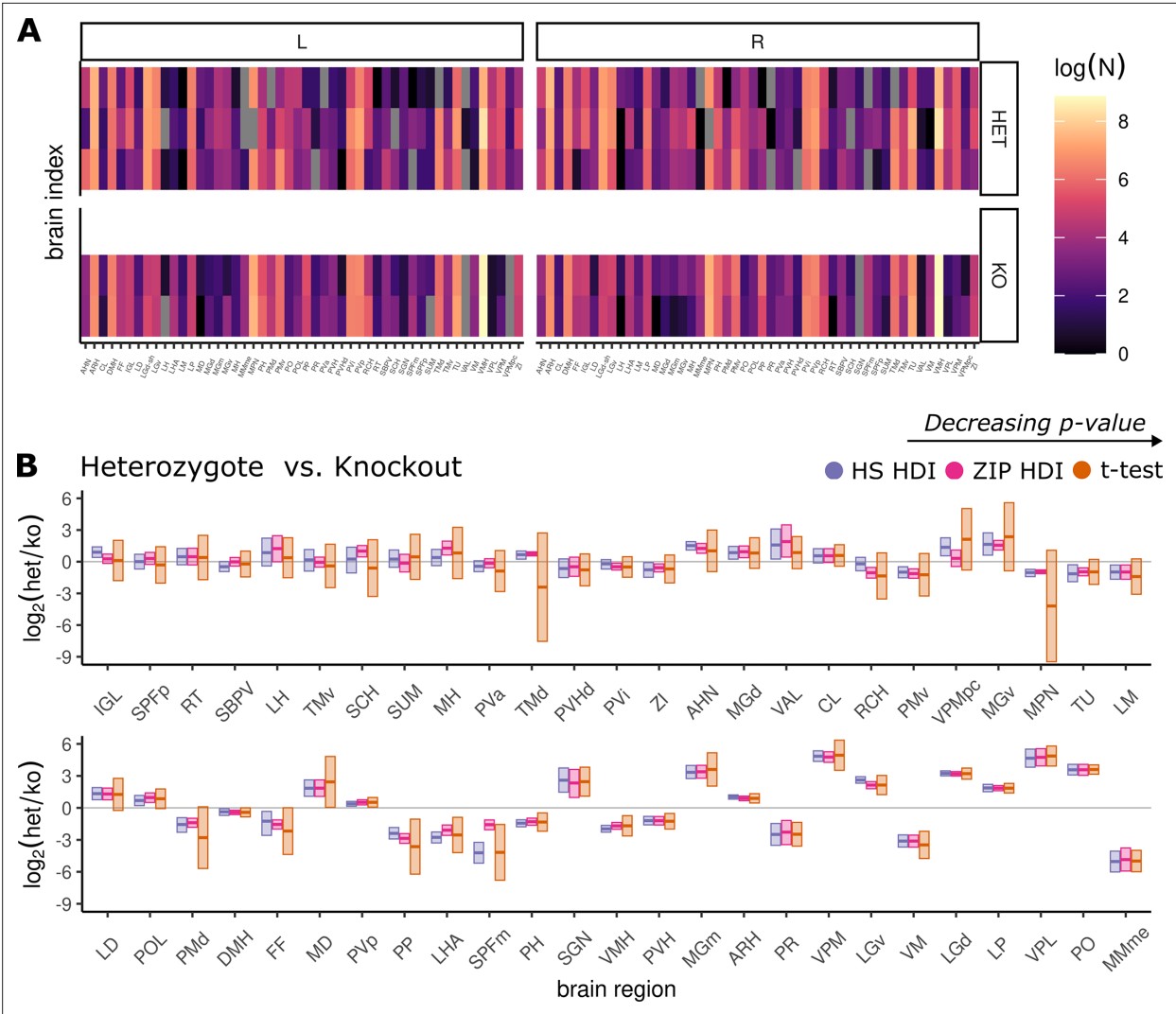

**Figure 5.** Results - Case study 2. (**A**) Heatmap of the raw log cell count data. Each row corresponds to a single animal, columns correspond to brain regions. L and R denote left and right hemispheres, respectively. (**B**) log₂ fold differences in green fluorescent protein (GFP) positive cells between mouse genotypes, heterozygous (HET), and knockout (KO), for each of the 50 recorded brain regions spread across two rows. The 95% Bayesian highest density interval (HDI) is given in purple and pink for the Bayesian horseshoe and zero-inflated model. The 95% confidence interval calculated from a Welch's *t*-test is in orange. Horizontal lines within the intervals mark the posterior mean of the Bayesian results and the data estimate for the *t*-test. The *x*-axis is ordered in terms of decreasing p-value from the significance test.

In contrast, the Bayesian estimates (green bars, *Figure 4*) produce a clear result. For a number of brain regions in *Figure 4A*, sham-novel animals have higher expression than sham-familiar ones. These differences disappear in *Figure 4B* with lesion-novel and lesion-familiar animals showing roughly equal cell counts. This indicates that the difference is only present when the NRe is intact.

## Case study 2 - Ontogeny of inhibitory interneurons of the mouse thalamus

For each of the 50 brain regions, the estimated $\log_2$-fold difference in GFP-expressing cells between the two genotypes is plotted in *Figure 5*. This includes the purple and pink 95% HDI from the horse-shoe and zero-inflated Poisson models along with the 95% CI arising from a $t$-test in orange. For most brain regions, the two Bayesian models gave narrower HDIs than the $t$-test CI. Accordingly, the Bayesian models identified a greater number of brain regions that had genotype differences in Sox14-positive cell count in the sense that they found more places where the appropriate uncertainty interval does not overlap zero.

Despite the large difference in interval estimation between the Bayesian HDIs and $t$-test CI for many brain regions, as the data becomes stronger from the perspective of the frequentist p-value toward the right-hand side of the second row in *Figure 5*, the model results become much more compatible. The variation within groups is very small for these regions. Further regularization is not necessary, and so the impact of partial pooling has been reduced. The sample estimate of the $t$-test has 'caught up' to the regularized estimate because the signal is strong.

The zero-inflated Poisson distribution sometimes differs from the $t$-test CIs. One example of this is the result for the dorsal tuberomammillary nucleus (TMd). *Figure 6*, bottom row, plots the raw cell-count values for TMd alongside the inferred frequentist mean and two Bayesian model means. For this region, the HET animals have high GFP expression across both hemispheres, yet animal three has a reading of zero for both hemispheres. This injects variability into the standard deviation of the HET group. Consequently, the pooled standard deviation used in the $t$-test is large and almost certainly guarantees a nonsignificant result. Furthermore, the sample mean of this region looks nothing like zero, but also nothing like the other two animals with positive counts. In addition to the wide interval, the sign of the difference does not agree with the data. The medial preoptic nucleus (MPN) also suffers from poor estimation. Once again, this region contains a single HET animal for which the reading from both hemispheres is zero. The zero-inflated Poisson produces a posterior distribution of the appropriate sign with small uncertainty.

The two Bayesian models did not always agree. In some cases, such as the medial habenula (MH) and suprachiasmatic nucleus (SCH), the 'standard' horseshoe model does not show a genotype difference in cell counts, while the ZIP model indicates that heterozygotes had higher cell counts than KO (*Figures 5 and 6*). The opposite can be seen in the case of the parvicellular ventral posteromedial nucleus of the thalamus (VPMpc), the horseshoe model suggests a genotype difference where the ZIP model did not (*Figure 5*). Further examination of the data shows why this happens, for example, for region MH (*Figure 6*, top row), the 'standard' horseshoe model sensibly ignores the large positive outlier value in the heterozygote data, while the ZIP model does not. As a result, the ZIP model's estimate for the mean is pulled upward, leading to an inferred difference in heterozygote versus the wild type.

## Discussion

We have presented a standard workflow for Bayesian analysis of multiregion cell-count data. We propose a likelihood and appropriate priors with a nested hierarchical structure reflecting the structure of the experiment. We applied this to two distinct example datasets and demonstrated that they capture more fruitfully the characteristics of the data when compared to field-standard frequentist analyses.

For both case studies, the Bayesian uncertainty intervals are more precise than the CIs. These CIs tend to be quite wide on these data because of the small sample size and because of violations to their parametric model assumptions.

Our standard workflow uses a horseshoe prior, along with the partial pooling. This allows our model to deal effectively with outliers. Furthermore, for the data sizes presented here, a full Bayesian

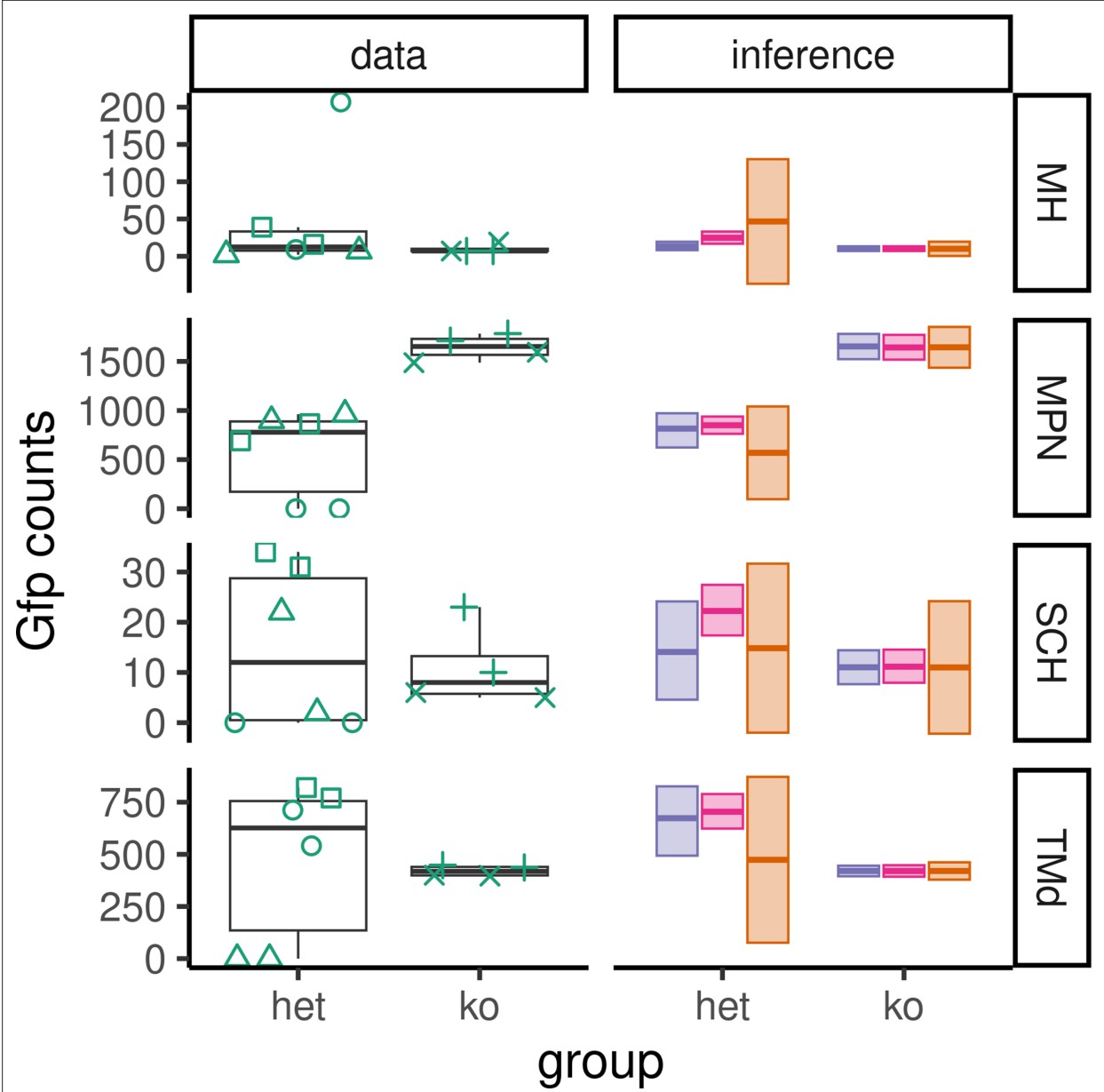

**Figure 6.** Example data and inferences highlighting model discrepancies. On the left under 'data': boxplots with medians and interquartile ranges for the raw data for four example brain regions. The shape of each point pairs left and right hemisphere readings in each of the five animals. On the right under 'inference': highest density intervals (HDIs) and confidence intervals are plotted. Purple is the Bayesian horseshoe model, pink is the Bayesian ZIP model, and orange is the sample mean. The Bayesian estimates are not strongly influenced by the zero-valued observations (medial preoptic nucleus [MPN], suprachiasmatic nucleus [SCH], dorsal tuberomammillary nucleus [TMd]) or large-valued outliers (medial habenula [MH]) and have means close to the data median. This explains the advantage of the Bayesian results over the confidence interval.

inference using Stan does not require long computation time, or even particularly high-performance hardware. Modern multicore laptop processors are quite sufficient for this task. Fitting a model typically takes less than an hour.

In our analysis, we have noted examples where different Bayesian models give discrepant conclusions. The obvious question to ask is, which should we trust? The disappointing but inevitable answer is that, as with more traditional methods, Bayesian analysis is only a tool useful for interpreting data and brings with it a set of assumptions and biases regarding the experiment and the data. A Bayesian analysis does not avoid inconsistent or inconclusive results, but it usually makes the assumptions more explicit and transparent. Typically, the solution to these model inconsistencies is to inspect the

raw data and ask which model better captures those aspects of the data we are most interested in. Overall, the lesson here is that Bayesian hierarchical modeling has greater flexibility and statistical power, but all statistical analyses, even those claiming to 'test hypotheses', just support exploration, and it is ultimately the researcher's responsibility to make sure that a model's assumptions are appropriate and its behavior is sensible for the target dataset.

The horseshoe prior model workflow we have exhibited here is intended as a standard approach. We believe that, without extension, it will provide a robust model for cell-count data. However, we also suggest that the standard workflow can be a useful first step for a more comprehensive, extended model when one is required. We have given an example of this for the second dataset where the anomalous zeros prompted us to change the likelihood to a zero-inflated Poisson. There are other possibilities, e.g., zero inflation is not the only way to handle an anomaly in the number of zeros: the hurdle model is an alternative (*Cragg, 1971*). This is not a mixture model; instead, it restricts the probability of zeros to some value $\pi$ with the probabilities for the positive counts coming from a truncated Poisson distribution. The hurdle model can deflate, as well as inflate, the probability mass at zero. This did not match the situation in the data we considered but might for other datasets. Another extension might involve tighter priors based on previous experiments. This is likely to be very relevant for cell-count data since these experiments are rarely performed in isolation, and so prior information can be leveraged from a history of empirical results.

One obvious elaboration of our model would replace normal distributions with multivariate normal distributions. This would have two advantages. First, correlations are difficult to estimate for under-sampled data. Including correlation matrix priors provides extra information - e.g., based on anatomical connectivity - that can aid the statistical estimation of other parameters. Second, it would more closely match our understanding of the experiment: we know that activity is likely to be correlated across regions, and so it is apposite to include that directly in the model. Unfortunately, the problem of finding a suitable prior for the correlation proved insurmountable: the standard Lewandowski-Kurowicka-Joe distribution (*Lewandowski et al., 2009*) which has been useful in lower-dimensional situations is too regularizing here. This is an area where further work needs to be done.

It is important to highlight that a mixed effects model is not a uniquely Bayesian construction. Indeed, any model that tries to include more sophistication through hierarchical structures, Bayesian or otherwise, is useful. However, non-Bayesian models can be complicated and opaque; they are also often more restrictive. For example, they often assume normal distributions, and circumventing these restrictions can make the models even less transparent. A Bayesian approach is, at first, unfamiliar; this can make it seem more obscure than better established methods, but, in the long run, Bayesian models are typically clearer and do not involve so many different assumptions and so many fine adjustments.

## Acknowledgements

We are grateful to Andrew Dowsey and Matthew Nolan for useful discussion and helpful suggestions.

## Additional information

### Funding

| Funder | Grant reference number | Author |
| --- | --- | --- |
| Engineering and Physical Sciences Research Council | EP/R513179/1 | Sydney Dimmock |
| Engineering and Physical Sciences Research Council | EP/W024020/1 | Simon R Schultz |
| Biotechnology and Biological Sciences Research Council | BB/L02134X/1 | E Clea Warburton |
| Biotechnology and Biological Sciences Research Council | BB/R007020/1 | Alessio Delogu |

| Funder | Grant reference number | Author |
| --- | --- | --- |
| Wellcome Trust | 206401/Z/17/Z | E Clea Warburton |
| Leverhulme Trust | RF-2021-533 | Conor J Houghton |
| Medical Research Council | MR/S026630/1 | Cian O'Donnell |

The funders had no role in study design, data collection and interpretation, or the decision to submit the work for publication. For the purpose of Open Access, the authors have applied a CC BY public copyright license to any Author Accepted Manuscript version arising from this submission.

## Author contributions

Sydney Dimmock, Conceptualization, Data curation, Software, Formal analysis, Investigation, Visualization, Methodology, Writing – original draft, Writing – review and editing; Benjamin MS Exley, Gerald Moore, Lucy Menage, Resources, Data curation; Alessio Delogu, Simon R Schultz, Resources, Funding acquisition, Writing – review and editing; E Clea Warburton, Resources, Supervision, Funding acquisition, Writing – review and editing; Conor J Houghton, Conceptualization, Formal analysis, Supervision, Funding acquisition, Investigation, Methodology, Writing – original draft, Writing – review and editing; Cian O'Donnell, Conceptualization, Supervision, Funding acquisition, Investigation, Methodology, Writing – original draft, Project administration, Writing – review and editing

## Author ORCIDs

Sydney Dimmock ⓘ https://orcid.org/0000-0002-0163-2048
Alessio Delogu ⓘ https://orcid.org/0000-0002-4414-4714
Simon R Schultz ⓘ https://orcid.org/0000-0002-6794-5813
E Clea Warburton ⓘ https://orcid.org/0000-0002-2129-2060
Conor J Houghton ⓘ https://orcid.org/0000-0001-5017-9473
Cian O'Donnell ⓘ https://orcid.org/0000-0003-2031-9177

Reviewer #1 (Public review): https://doi.org/10.7554/eLife.102391.3.sa1
Author response https://doi.org/10.7554/eLife.102391.3.sa2

---

# Additional files

## Supplementary files

MDAR checklist

## Data availability

The code necessary to run the models presented in this manuscript can be found at *Dimmock et al., 2025* and on our Github https://BayesianCellCounts.github.io. The data for case study one on nucleus reuniens lesion are available from https://doi.org/10.5281/zenodo.12787211 (*Exley et al., 2024*). The data from case study two on Sox14 expressing neurons are available from https://doi.org/10.5281/zenodo.12787287 (*Gerald and Sydney, 2024*).

---

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

## Appendix 1

### Full model

Here, we present the complete mathematical model for each of the three models applied in the main text. In all cases, the exposure term is only necessary for the first case study. The area was not available for the second, so the exposure term was left out.

#### Poisson model

$$y_i \sim \text{Poisson}(\lambda_i) \tag{1}$$

$$\log \lambda_i = \theta_{r[i],g[i]} + E_i + \gamma_i \tag{2}$$

$$\theta_{rg} \sim \text{Normal}(5, 2) \tag{3}$$

$$\gamma_i \sim \text{Normal}\left(0, \tau_{r[i],g[i]}\right) \tag{4}$$

$$\tau_{rg} \sim \text{HalfNormal}(\log(1.05)) \tag{5}$$

#### Horseshoe model

$$y_i \sim \text{Poisson}(\lambda_i) \tag{6}$$

$$\log \lambda_i = \theta_{r[i],g[i]} + E_i + \gamma_i \tag{7}$$

$$\theta_{rg} \sim \text{Normal}(5, 2) \tag{8}$$

$$\gamma_i \sim \text{Normal}\left(0, \kappa_i \times \tau_{r[i],g[i]}\right) \tag{9}$$

$$\tau_{rg} \sim \text{HalfNormal}(\log(1.05)) \tag{10}$$

$$\kappa_i \sim \text{HalfNormal}(1) \tag{11}$$

#### Zero-inflated model

$$y_i \sim \text{ZIPoisson}(\lambda_i, \pi) \tag{12}$$

$$\pi \sim \text{Beta}(1, 5) \tag{13}$$

$$\log \lambda_i = \theta_{r[i],g[i]} + E_i + \gamma_i \tag{14}$$

$$\theta_{rg} \sim \text{Normal}(5, 2) \tag{15}$$

$$\gamma_i \sim \text{Normal}\left(0, \tau_{r[i],g[i]}\right) \tag{16}$$

$$\tau_{rg} \sim \text{HalfNormal}(\log(1.05)) \tag{17}$$

### Distributions

#### Zero-inflated Poisson distribution

The zero-inflated Poisson distribution is a mixture distribution with mixing parameter $\pi$. The distribution is formally defined below. If

$$p_Y(y) \equiv \text{ZIPoisson}(\pi, \lambda) \tag{18}$$

$$p_Y(y) = \begin{cases} \pi + (1 - \pi) \exp(-\lambda) & \text{if} \quad y = 0 \\ (1 - \pi)f(y) & \text{otherwise} \end{cases} \tag{19}$$

where

$$f(y) \equiv \text{Poisson}(\lambda) \tag{20}$$

Equivalently, the mixture can be specified with an indicator function.

$$p_Y(y) = \mathbf{1}_0(y)\pi + (1 - \pi)f(y) \tag{21}$$

## HalfNormal distribution

The HalfNormal distribution coincides with a zero-mean normal distribution truncated at zero. It has a single scale parameter $\sigma$. If

$$p_Y(y) \equiv \text{Half Normal}(\sigma) \tag{22}$$

then

$$p_Y(y) = \frac{\sqrt{2}}{\sigma\pi} \exp\left(-\frac{y^2}{2\sigma^2}\right) \tag{23}$$

is the probability density function.

## **Additional methods**

### Fold differences

In our results, we present differences between experimental groups in terms of $\log_2$-fold differences. We calculate this as follows. The parameter of interest $\theta_{r,g=i}$ is modeled on the natural log scale owing to the log-link function necessary for the Poisson regression. At the average animal, the difference

$$\theta_{r,g=i} - \theta_{r,g=j} = \log \frac{\exp(\theta_{r,g=i})}{\exp(\theta_{r,g=j})} \tag{24}$$

$$= \log \Delta_{ij} \tag{25}$$

is the natural log of the ratio of the expected counts. To obtain $\log_2$-fold differences, we simply change the base by multiplying $\log \Delta_{ij}$ by $\log_2(e)$.

### Data transformations

To facilitate a comparison with the Bayesian intervals in terms of $\log_2$-fold differences, it was necessary to add one to any zero counts before applying the $t$-test.

### Non-centered parameterization

Hierarchical models can produce geometry that is difficult for the sampler to explore. Fortunately, there exists a simple reparameterization known as non-centering that can remedy this problem. In our model, instead of sampling $\gamma_i$ directly, we sample the parameter $\tilde{\gamma}_i$ instead and use it to reconstruct $\gamma_i$. That is, sample $\tilde{\gamma}_i$ from a standard normal distribution,

$$\tilde{\gamma}_i \sim \text{Normal}(0, 1) \tag{26}$$

and reconstruct $\gamma_i$ as a deterministic function of sampled values of $\tilde{\gamma}_i$ and $\tau_{r[i],g[i]}$.

$$\gamma_i = \theta_{r[i],g[i]} + \tau_{r[i],g[i]} \times \tilde{\gamma}_i \tag{27}$$

This removes the frustrating joint behavior between $\gamma_i$ and $\tau_{r[i],g[i]}$, and promotes efficient sampling.

### Preprocessing - Case study 1

In these data, some animals produced more than one reading per brain region. Before fitting our model, these were summed together to produce a single count; the exposure term was also properly adjusted to correctly reflect the area of the recording site.

## Software, packages, and libraries

**Appendix 1—table 1.** Software packages used.

| R Libraries | Version | Description |
| --- | --- | --- |
| rstan | 2.26.3 | complete Stan library |
| cmdstanr | 0.5.2 | lightweight Stan library |
| HDInterval | 0.2.2 | calculating HDI in R |
| ggplot2 | 3.4.1 | plotting |
| bayesplot | 1.9.0 | plotting |
| tidyverse | 1.3.1 | tibble, tidyr, readr, purr, dplyr, stringr, forcats |

R version 4.2.1 - 'Funny-looking-kid'.
Computation was performed locally on a Dell XPS 13 7390 laptop. Intel i7-10510U @ 1.80 GHz, 16 GB of RAM, Ubuntu 20.04.4 LTS.
Panels composed using Inkscape version 1.2.2.

**Appendix 1—table 2.** Acronyms for the brain regions in Case study 1.

| Term | Definition |
| --- | --- |
| ACC | Anterior cingulate cortex |
| DCA1/3 | Dorsal CA1/3 |
| DDG | Dorsal dentate gyrus |
| DPC | Dorsal peduncular cortex |
| DSUB | Dorsal subiculum |
| HPC | Hippocampus |
| ICA1/3 | Intermediate CA1/3 |
| IDG | Intermediate dentate gyrus |
| IFC | Infralimbic cortex |
| LENT | Lateral entorhinal cortex |
| MOC | Medial orbital cortex |
| MPFC | Medial prefrontal cortex |
| M2C | Motor cortex M2 |
| NRe | Nucleus reuniens |
| PRL | Prelimbic cortex |
| PRH | Perirhinal cortex |
| PSTC | Postrhinal cortex |
| TE2 | Temporal association cortex |
| VCA1/3 | Ventral CA1/3 |
| VDG | Ventral dentate gyrus |
| VOC | Ventral orbital cortex |
| VSUB | Ventral subiculum |

*Appendix 1—table 2 Continued on next page*

*Appendix 1—table 2 Continued*

| Term | Definition |
|------|-----------|
| V2C | Visual cortex V2 |

**Appendix 1—table 3.** Acronyms for the brain regions in Case study 2.

| Term | Definition | Term | Definition |
|------|-----------|------|-----------|
| AHN | Anterior hypothalamic nucleus | PP | Peripeduncular nucleus |
| ARH | Arcuate hypothalamic nucleus | PR | Perireunensis nucleus |
| CL | Central lateral nucleus of the thalamus | PVa | Periventricular hypothalamic nucleus, anterior part |
| DMH | Dorsomedial nucleus of the hypothalamus | PVH | Paraventricular hypothalamic nucleus |
| FF | Fields of Forel | PVHd | Paraventricular hypothalamic nucleus, descending division |
| IGL | Intergeniculate leaflet of the lateral geniculate complex | PVi | Periventricular hypothalamic nucleus, intermediate part |
| LD | Lateral dorsal nucleus of thalamus | PVp | Periventricular hypothalamic nucleus, posterior part |
| LM | Lateral mammillary nucleus | RCH | Retrochiasmatic area |
| LGv | Ventral part of the lateral geniculate complex | RT | Reticular nucleus of the thalamus |
| LGd | Dorsal part of the lateral geniculate complex | SBPV | Subparaventricular zone |
| LH | Lateral habenula | SCH | Suprachiasmatic nucleus |
| LHA | Lateral hypothalamic area | SGN | Suprageniculate nucleus |
| LP | Lateral posterior nucleus of the thalamus | SPFm | Subparafascicular nucleus, magnocellular part |
| MD | Mediodorsal nucleus of thalamus | SPFp | Subparafascicular nucleus, parvicellular part |
| MGd | Medial geniculate complex, dorsal part | SUM | Supramammillary nucleus |
| MGv | Medial geniculate complex, ventral part | TMd | Tuberomammillary nucleus, dorsal part |
| MGm | Medial geniculate complex, medial part | TMv | Tuberomammillary nucleus, ventral part |
| MH | Medial habenula | TU | Tuberal nucleus |
| MMme | Medial mammillary nucleus, median part | VAL | Ventral anterior-lateral complex of the thalamus |
| MPN | Medial preoptic nucleus | VMH | Ventromedial hypothalamic nucleus |
| PH | Posterior hypothalamic nucleus | VM | Ventral medial nucleus of the thalamus |
| PMd | Dorsal premammillary nucleus | VPL | Ventral posterolateral nucleus of the thalamus |
| PMv | Ventral premammillary nucleus | VPM | Ventral posteromedial nucleus of the thalamus |
| PO | Posterior complex of the thalamus | VPMpc | Ventral posteromedial nucleus of the thalamus, parvicellular part |
| POL | Posterior limiting nucleus of the thalamus | ZI | Zona incerta |

## Brain-region name acronyms

### Sampler diagnostics

The basic Poisson model was sampled excellently. Measures of sampling performance such as $\hat{R}$ (*Gelman and Rubin, 1992*; *Vehtari et al., 2021*) and effective sample size were all satisfactory (*Appendix 1—figure 1*). Similarly, for the zero-inflated model, no problems were observed for any of the diagnostics (*Appendix 1—figure 2*). Contrasting with this, the horseshoe model exhibited some signs of fitting problems (*Appendix 1—figure 3*). Divergences were not observed, and given the longer chain length, this is reassuring evidence against biased computation. However, for many parameters, the effective sample size is much lower than we would like to see. This is reflected in the trace plots: the sampler is not making large jumps across the parameter space, implying high autocorrelation and low effective sample size. Unfortunately, the horseshoe is notoriously hard to fit, and we resort to brute-force methods, such as increasing the number of iterations and reducing the step size of the sampler to improve the inference. In the following three plots, diagnostics have been summarized with the following three items:

- **A:** The performance of the sampler is illustrated by plotting $\hat{R}$ (R-hat, $\hat{R} \approx 1$ ideal) against the ratio of the effective number of samples (larger is better) for each parameter in the model. Points represent individual parameters in the model and have further been color-coded by their type. For example, all $\theta_{r,g}$ are colored in green. Points have also been scaled based on how numerous the parameters are, so the more numerous parameters have smaller dots, the less numerous, larger.
- **B:** A histogram comparing the marginal energy distribution $\pi_E$ and the transitional energy distribution $\pi_{\Delta E}$ of the Hamiltonian. Ideally, these distributions should match each other closely if the posterior distribution has been properly explored by the sampler.
- **C:** For each parameter type, the parameter with the 'poorest' mixing (largest $\hat{R}$) are presented with a post-warmup trace plot that overlays the ordered sequence of samples from each of the four chains. Corresponding points in A are marked with a black border and zero transparency.

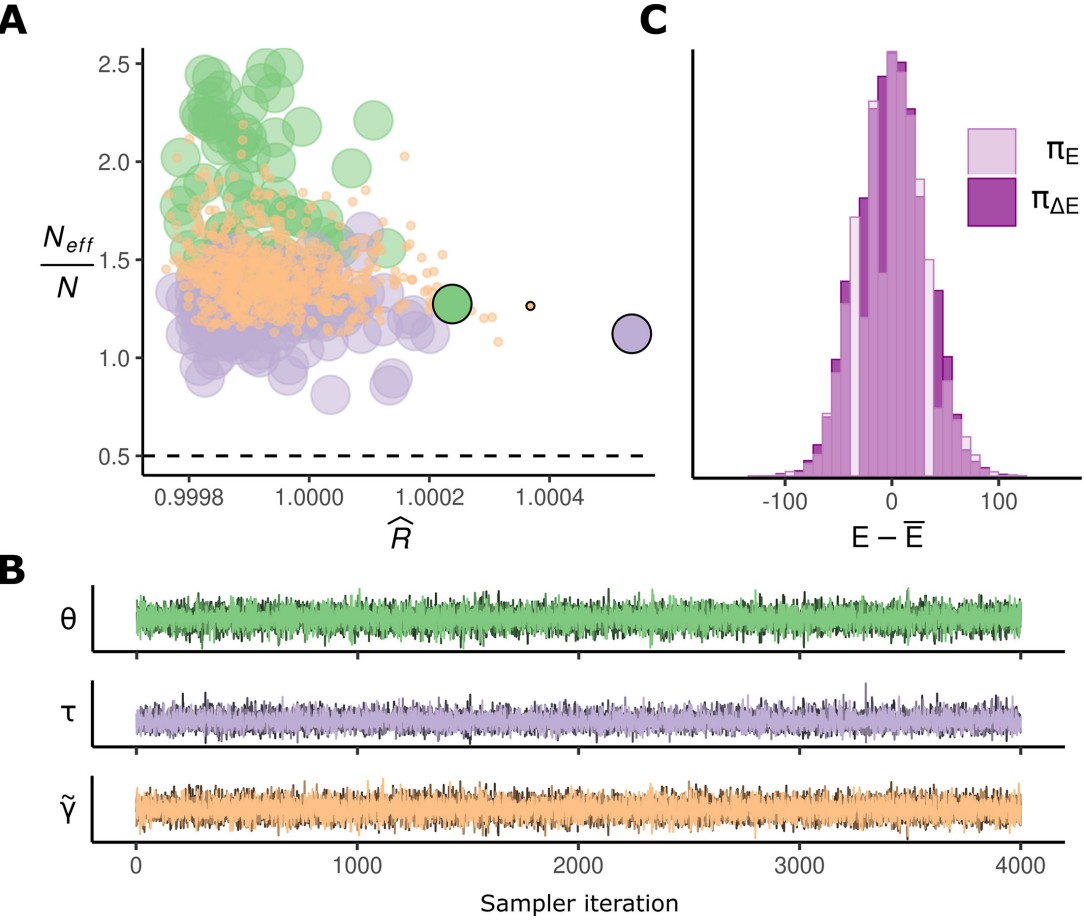

**Appendix 1—figure 1.** Diagnostics - Poisson. Standard Poisson model - Case study 1.

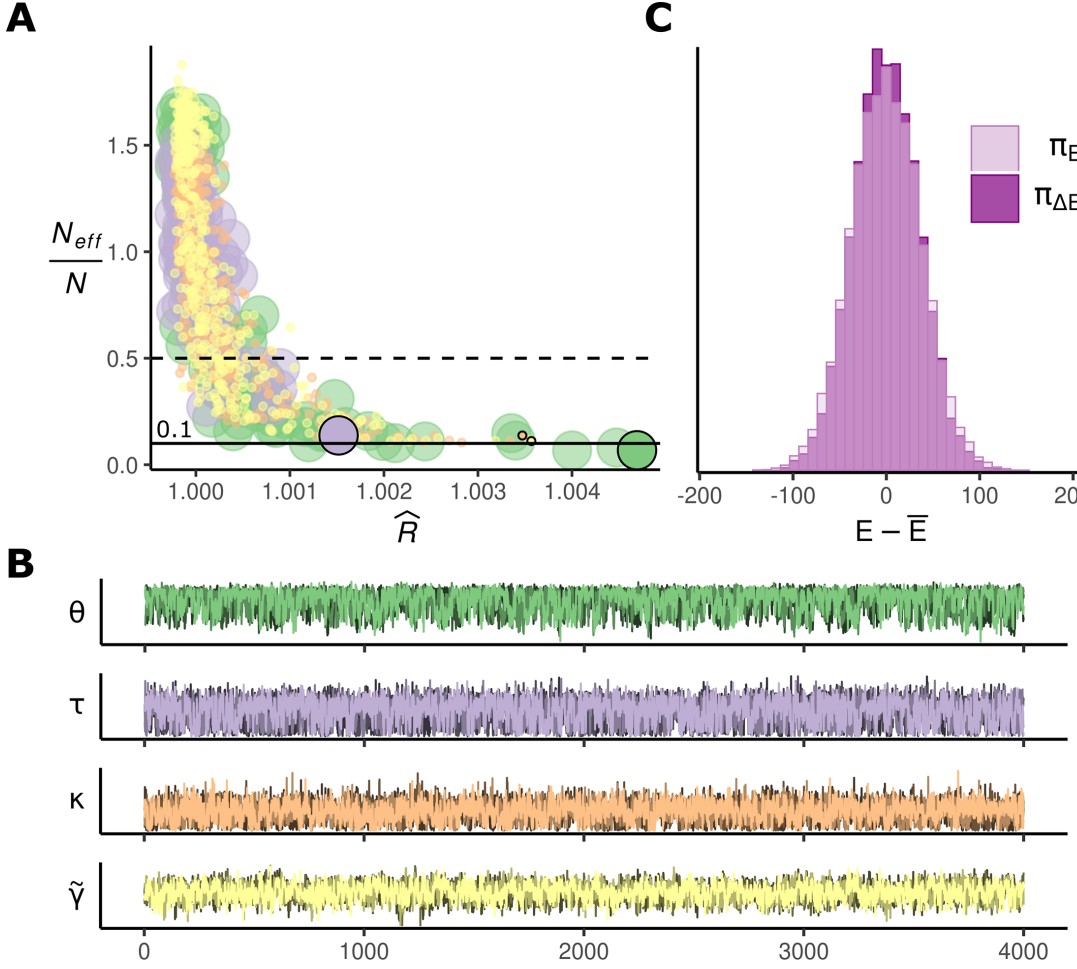

**Appendix 1—figure 2.** Diagnostics - Horseshoe. Horseshoe model - Case study 2.

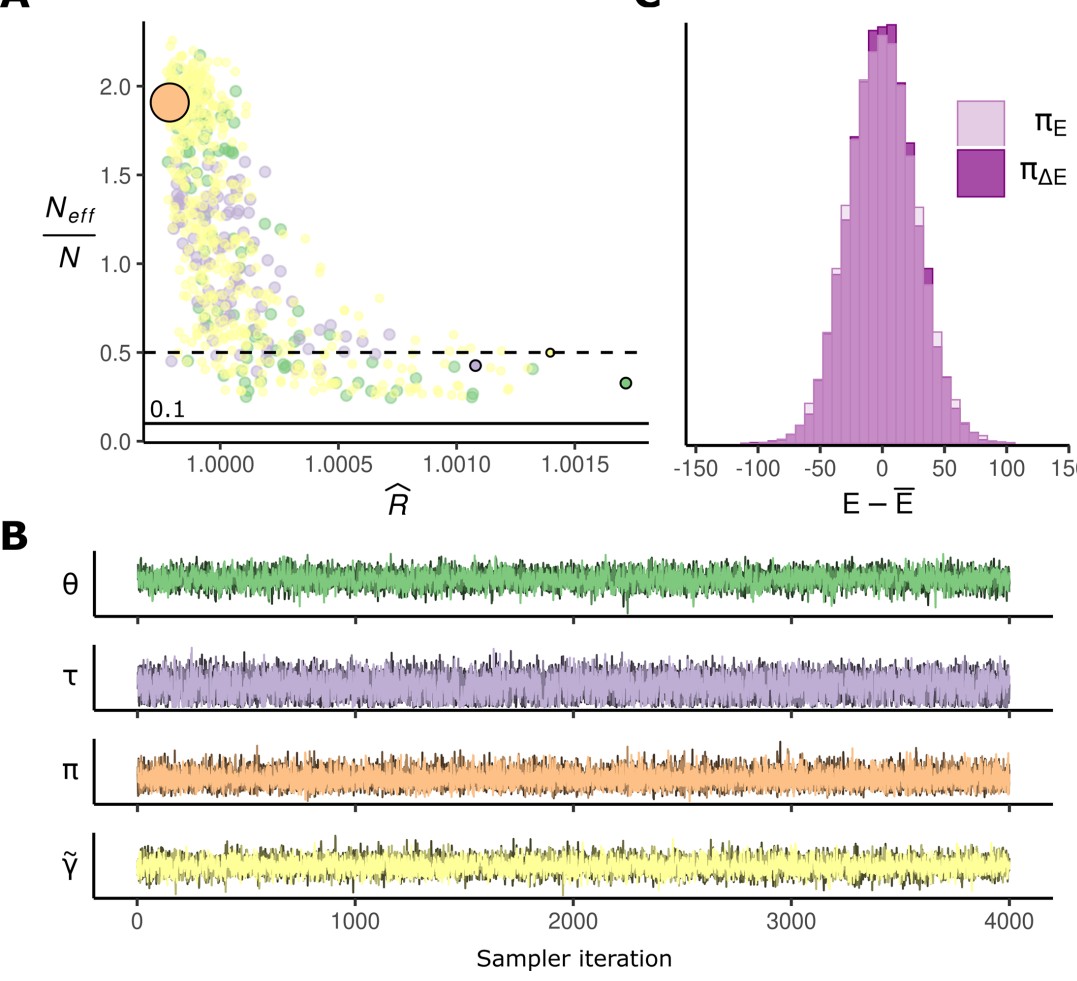

**Appendix 1—figure 3.** Diagnostics - ZIPoisson. Zero-inflated Poisson - Case study 2.

## Posterior predictive checking

Posterior predictive checks use the posterior predictive distribution,

$$p(y^{\text{rep}}|y) = \int p(y^{\text{rep}}|\theta)p(\theta|y)\, d\theta \tag{28}$$

where $p(\theta|y)$ is the posterior distribution and $p(y^{\text{rep}}|\theta)$ is the data distribution for $y^{\text{rep}}$ that follows the same form as the likelihood for $y$. If we can verify that the posterior predictive distribution can generate replicate datasets with similar statistics to the observed data, then we might conclude that our model is consistent with the observed data and useful for answering questions about it. In practice, a Monte Carlo approach is used to approximate statistics of the posterior predictive distribution. For example, if $T$ is the test statistic of interest such as the sample mean, then

1. For $1 \ldots S$,
   a. sample $\theta_s \backsim p(\theta|y)$
   b. sample $y_s^{\text{rep}} \sim p\left(y^{\text{rep}} \mid \theta_s\right)$
   c. calculate $T(y_s^{\text{rep}})$ where $T$ is the statistic of interest
2. Return $\{T(y_1^{\text{rep}}), \ldots, T(y_S^{\text{rep}})\} \approx p(T(y^{\text{rep}})|y)$

The posterior predictive checks that follow examine two important statistics of count data (*Appendix 1—figures 4–6*). (1) The standard deviation of the data (dispersion), as panel A. (2) The proportion of zeroes in the data (zero inflation), as panel B. The value of the test statistic applied

to the observed data $T(y)$ is plotted as a solid purple line, and the distribution of test statistics $\{T(y_1^{\mathrm{rep}}), \ldots, T(y_S^{\mathrm{rep}})\}$ as a purple histogram.

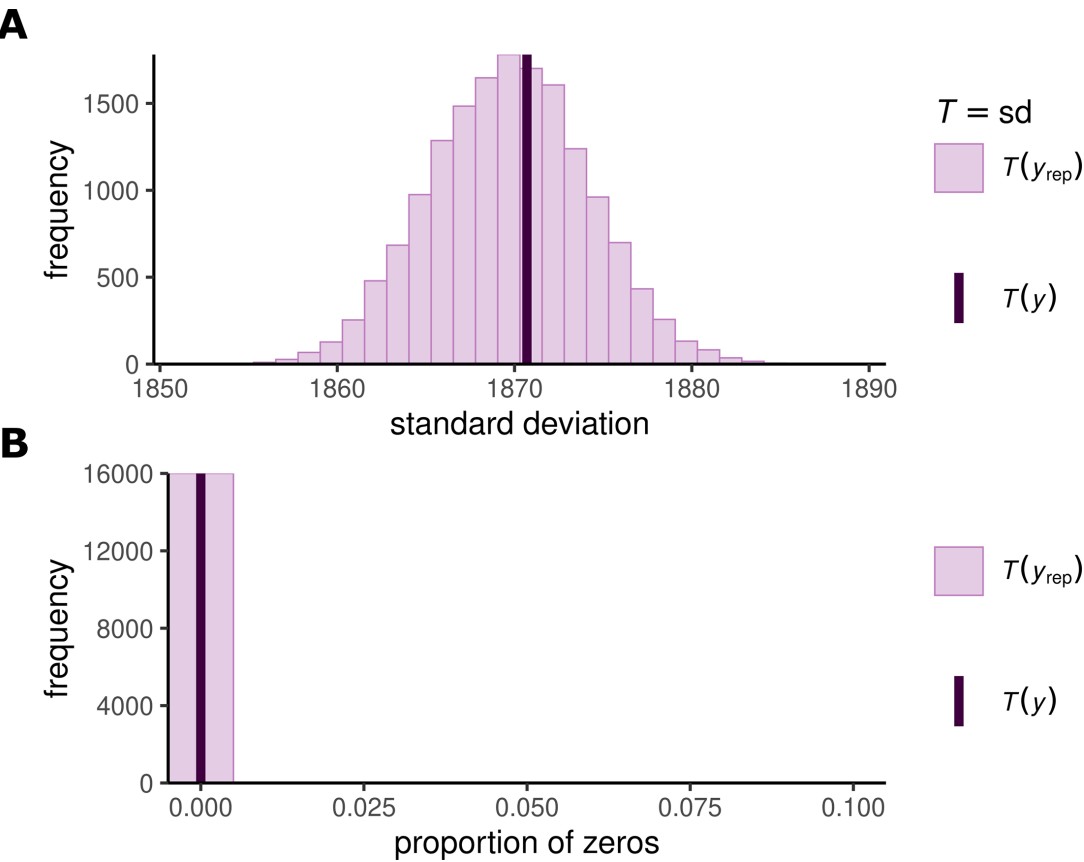

**Appendix 1—figure 4.** PPC - Poisson. Posterior predictive check for the standard Poisson model in Case study 1. (**A**) The proportion of zeroes in the data matches the proportion of zeroes in posterior predictive samples. This proportion is zero. (**B**) The distribution of standard deviations computed over a number of posterior predictive datasets (histogram) aligns with the standard deviation of the data.

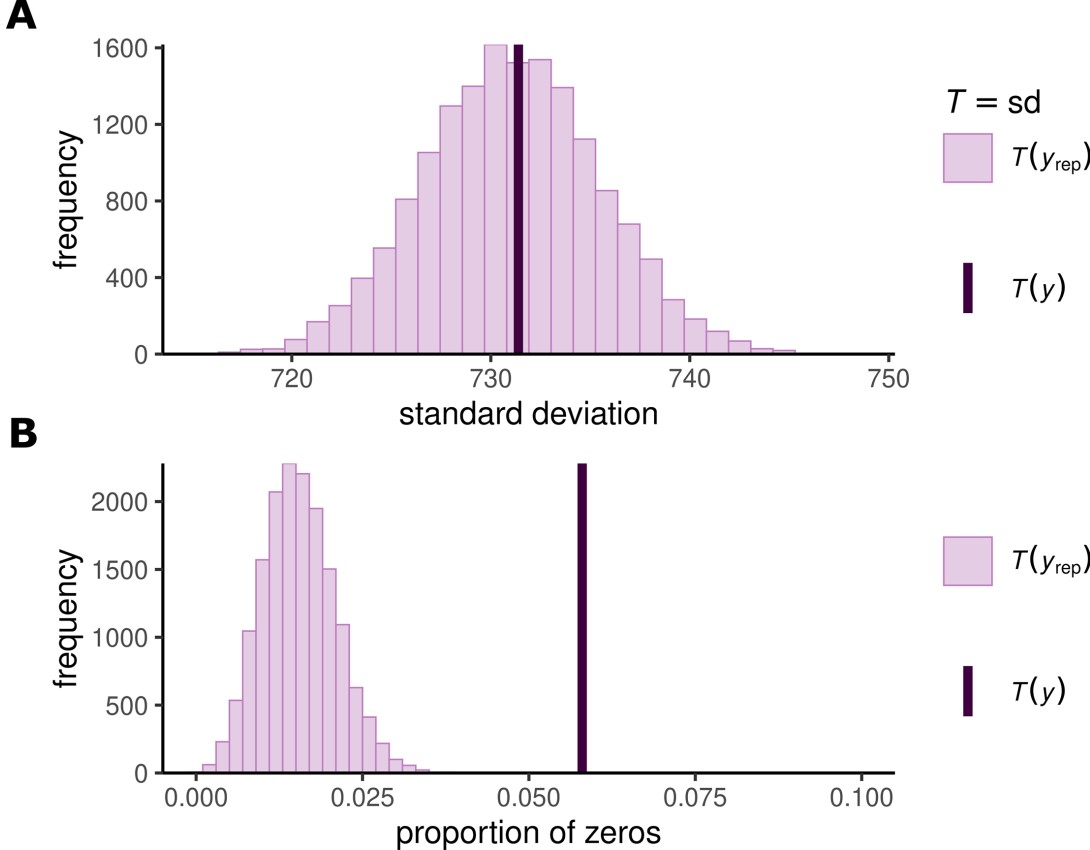

**Appendix 1—figure 5.** PPC - Horseshoe. Horseshoe model - Case study 2. Posterior predictive check for the standard horseshoe model in Case study 2. (**A**) The proportion of zeroes in the data is larger than those found in posterior predictive datasets. This makes sense, because the likelihood is still a Poisson distribution. (**B**) The distribution of standard deviations computed over a number of posterior predictive datasets (histogram) aligns with the standard deviation of the data.

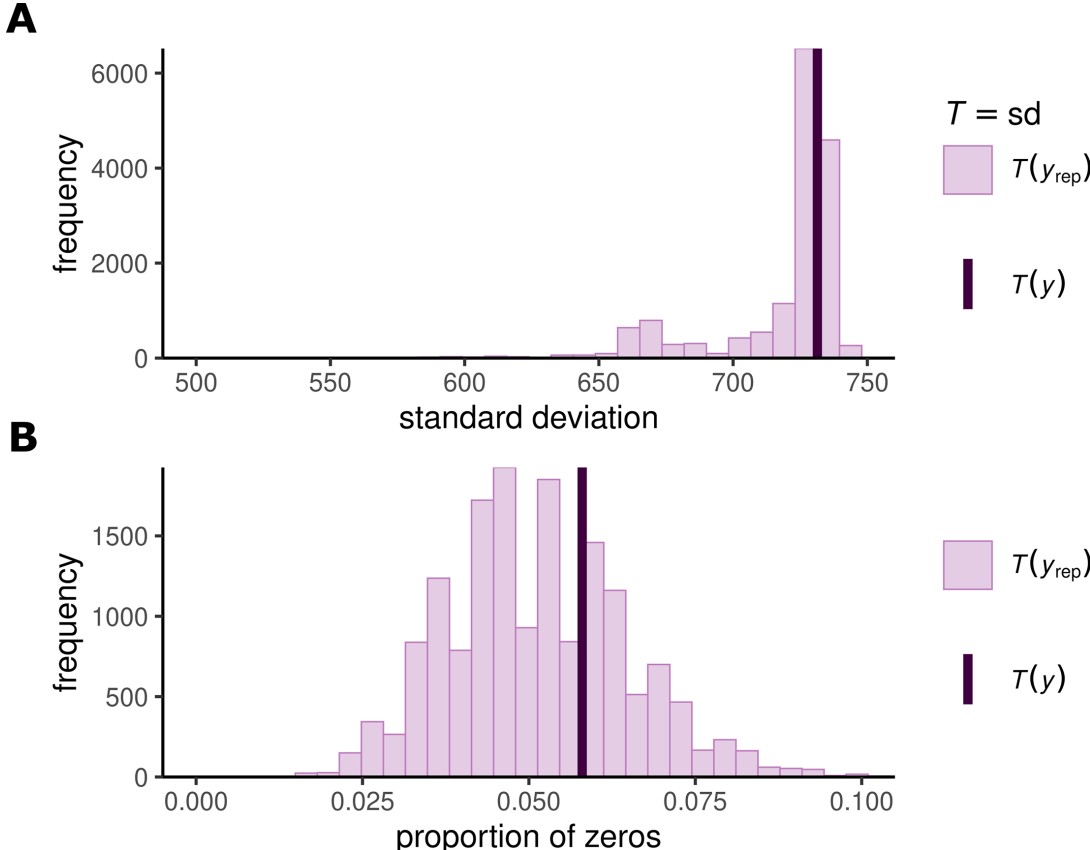

**Appendix 1—figure 6.** PPC - ZIPoisson. Zero-inflated Poisson - Case study 2. (**A**) The proportion of zeroes in the data matches the proportion of zeroes in posterior predictive samples. (**B**) The distribution of standard deviations computed over a number of posterior predictive datasets (histogram) aligns with the standard deviation of the data.

## Horseshoe densities

In our model, a horseshoe prior was used to allow some $\gamma_i$, typically those informed by $y_i = 0$, to escape regularization by partial pooling. However, we encountered many problems with the default parameterization that assigns a HalfCauchy density to the individual inflation parameters $\kappa_i$. In *Appendix 1—figure 7A*, the proportional conditional posterior density

$$p(\tilde{\gamma}_i, \kappa_i | \theta, \tau, y_i) \propto p(y_i | \tilde{\gamma}_i, \theta, \tau, \kappa_i) p(\tilde{\gamma}_i) p(\kappa_i) \tag{29}$$

where $\theta$ and $\tau$ have been fixed, is plotted when $y_i$ lies close to the population mean (left) or when it equals zero (right). Note that the x-axis is $\tilde{\gamma}_i$ and not $\gamma_i$ because the model samples a non-centered parameterization. That is,

$$\tilde{\gamma}_i = \frac{\gamma_i - \theta}{\tau \kappa_i}, \quad \tilde{\gamma}_i \sim \text{Normal}(0, 1) \tag{30}$$

captures the deviations of $\gamma_i$ around the population mean $\theta$.

*Appendix 1—figure 7B* plots samples from the marginal posterior $p(\tilde{\gamma}, \kappa_i | y_i)$, when fitting to the data $\mathbf{y} = \{770, 820, 713, 541, 0, 0\}$. Each data point in *Table 1* corresponds to one of the six plots in *Appendix 1—figure 7B*. This small example dataset is, in fact, the cell counts for region TMd recorded from the heterozygote group in Case study 2. The similarities in geometry can be readily seen with *Appendix 1—figure 7A*. A large number of divergences were produced by the sampler as demonstrated by the number of pink points compared to the non-divergent transitions in blue. For fixed $\tau$ the values $\tilde{\gamma}$ can take increase or decrease with smaller or larger $\kappa$, respectively. The

Half-Cauchy places an extremely long right tail over $\kappa$ that frustrates this relationship, resulting in a posterior density that is difficult to sample from.

## Modified horseshoe

For a Poisson model with parameters modeled on the log scale, we consider the Cauchy parameterization to be too extreme. In light of this, we opted for a pragmatic approach, a modification to the original horseshoe by replacing the HalfCauchy distribution with a HalfNormal distribution. The modified horseshoe cuts off the top of the funnels by restricting $\kappa$ to produce pleasant posterior geometry (*Appendix 1—figure 7A and B*). The modified horseshoe is much easier to sample from, but with the cost of a much more constraining prior over $\gamma_i$.

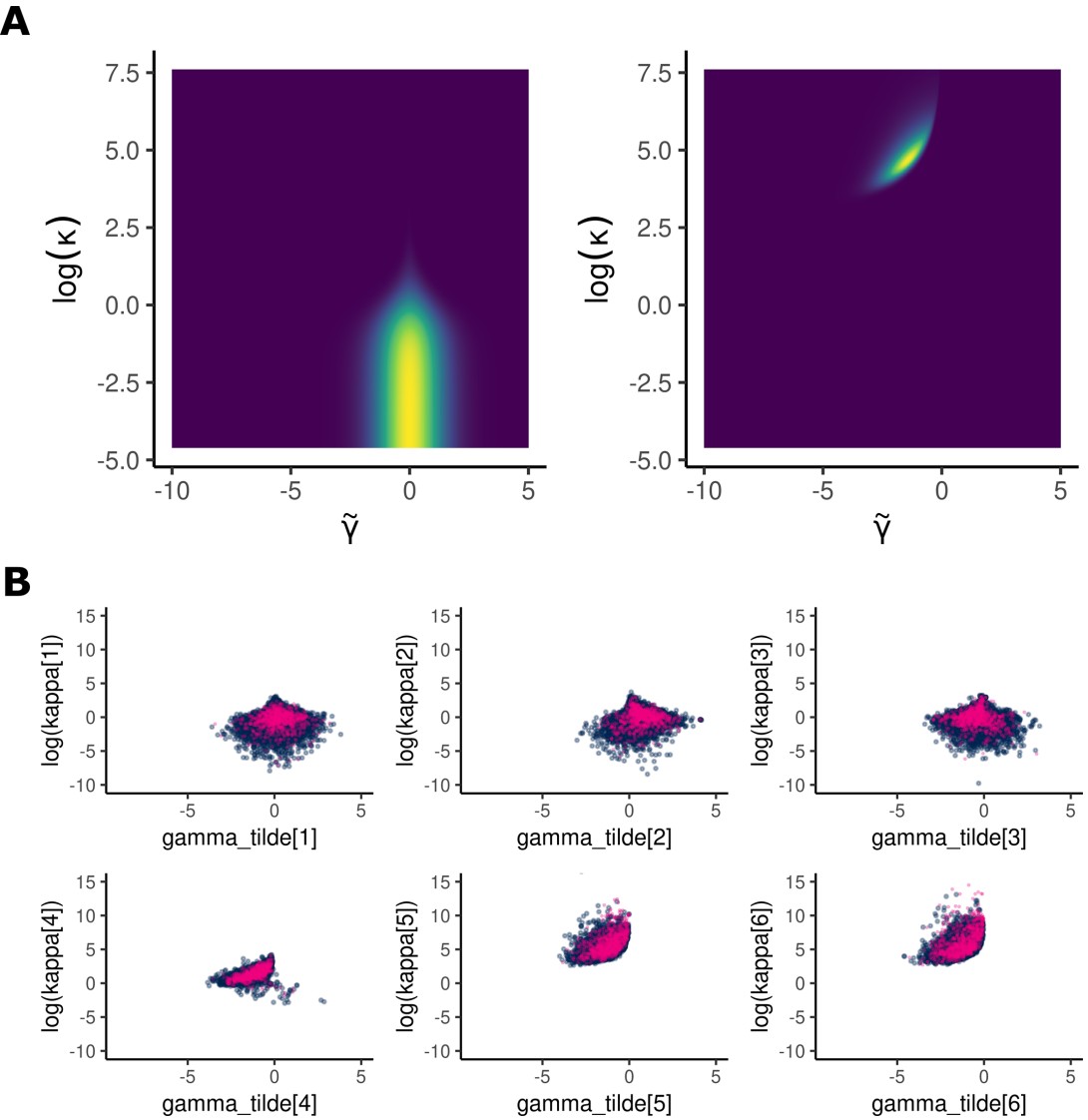

**Appendix 1—figure 7.** Horseshoe densities. (**A**) Conditional posterior. (**B**) MCMC pair plots. Divergent samples are colored in pink, non-divergent in blue.

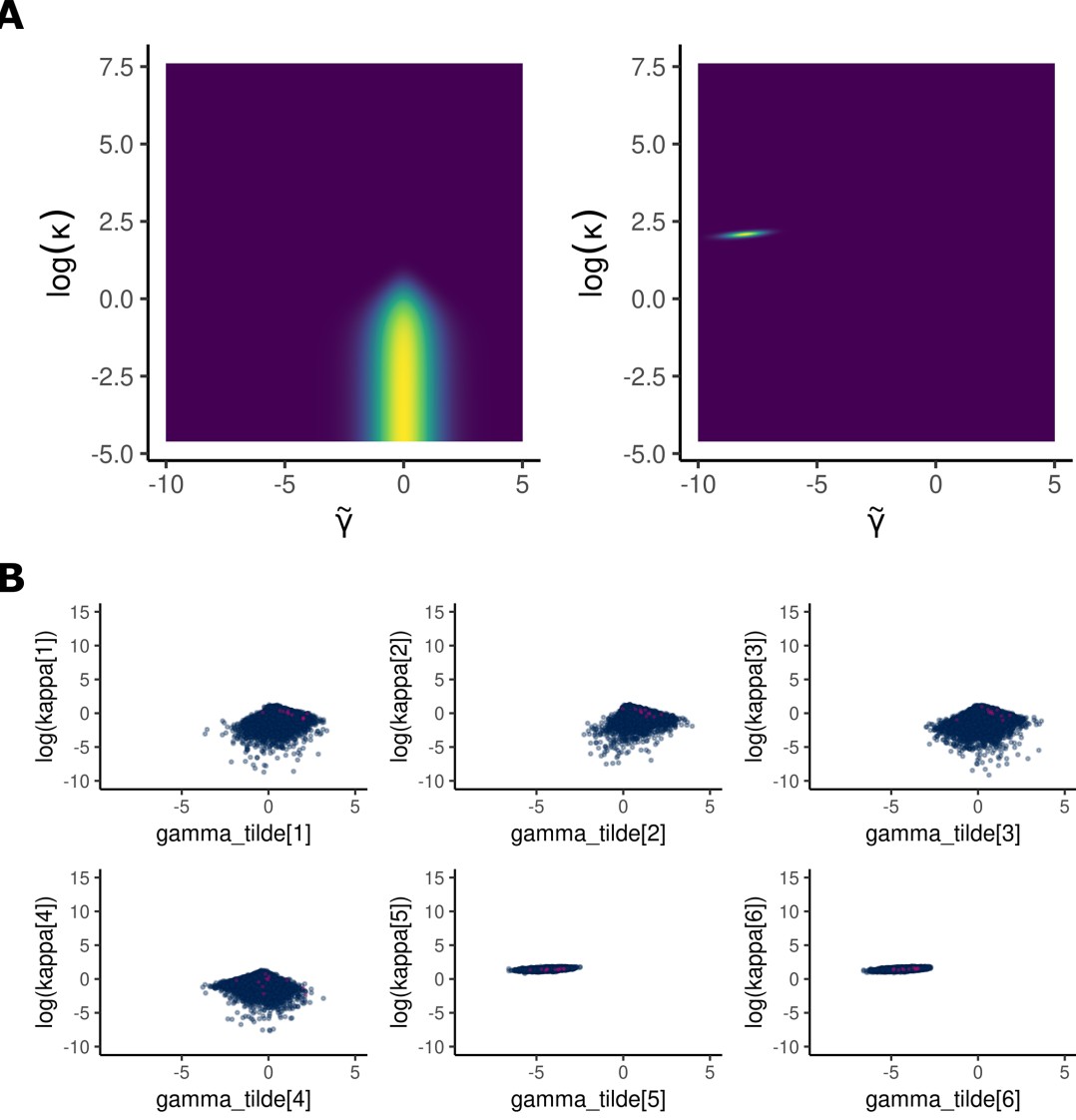

**Appendix 1—figure 8.** Modified horseshoe densities. (**A**) The conditional posterior $p(\tilde{\gamma}, \kappa \mid \theta, \tau, y)$ when $y = 0$ (left) and $y \neq 0$ (right). (**B**) MCMC pair plots of samples from the marginal posterior density $p(\tilde{\gamma}, \kappa \mid \mathbf{y})$.

