## [Editor Report · eLife Assessment]

This study proposes an **important** new approach to analyzing cell-count data, which are often undersampled and cannot be accurately assessed using traditional statistical methods. The case studies presented in the article provide **compelling** evidence of the superiority of the proposed methodology over existing approaches, which could promote the use of Bayesian statistics among neuroscientists. The authors have taken steps to make the methodology accessible, although some implementation difficulties are likely to remain.

---

## [Referee Report · Reviewer #1 (Public review)]

Summary:

This work proposes a new approach to analyse cell-count data from multiple brain regions. Collecting such data can be expensive and time-intensive, so, more often than not, the dimensionality of the data is larger than the number of samples. The authors argue that Bayesian methods are much better suited to correctly analyse such data compared to classical (frequentist) statistical methods. They define a hierarchical structure, partial pooling, in which each observation contributes to the population estimate to more accurately explain the variance in the data. They present two case studies in which their method proves more sensitive in identifying regions where there are significant differences between conditions, which otherwise would be hidden.

Strengths:

The model is presented clearly, and the advantages of the hierarchical structure are strongly justified. Two alternative ways are presented to account for the presence of zero counts. The first involves the use of a horseshoe prior, which is the more flexible option, while the second involves a modified Poisson likelihood, which is better suited to datasets with a large number of zero counts, perhaps due to experimental artifacts. The results show a clear advantage of the Bayesian method for both case studies.

The code is freely available, and it does not require a high-performance cluster to execute for smaller datasets. As Bayesian statistical methods become more accessible in various scientific fields, the whole scientific community will benefit from the transition away from p-values. Hierarchical Bayesian models are an especially useful tool that can be applied to many different experimental designs. However, while conceptually intuitive, their implementation can be difficult. The authors provide a good framework with room for improvement.

Weaknesses:

As with any Bayesian model, the choice of prior can significantly influence the results. The authors explain how the methodology can be adapted to different data properties, though selecting an appropriate prior or likelihood may not always be straightforward. They propose a 'standard workflow' as an alternative to traditional approaches, which could and should be used alongside established methods while Bayesian techniques continue to evolve and improve.

---

## [Author Response]

The following is the authors’ response to the original reviews.

**Reviewer #1 (Public review):**
“Alternative possibilities are discussed regarding the prior and likelihood of the model. Given that the second case study inspired the introduction of the zero-inflation likelihood, it is not clear how applicable the general methodology is to various datasets. If every unique dataset requires a tailored prior or likelihood to produce the best results, the methodology will not easily replace more traditional statistical analyses that can be applied in a straightforward manner. Furthermore, the differences between the results produced by the two Bayesian models in case study 2 are not discussed. In specific regions, the models provide conflicting results (e.g., regions MH, VPMpc, RCH, SCH, etc.), which are not addressed by the authors. A third case study would have provided further evidence for the generalizability of the methodology.”

We hope in this paper to propose a ‘standard workflow’ for these data; this standard workflow uses the horseshoe prior and we propose that this is the approach used to describe cell count data instead of the better established, but to our thinking, inefficient, t-testing approach.

The horseshoe prior is robust and allows a partially-pooled model to used while weighing-up the contribution of different data points. This is an analogue of excluding outliers and, in any analysis it is normal to investigate further if there are points being excluded as outliers. Often, this reveals a particular challenge with the data, in the case of the data here, there are a lot of zeros, indicating that some samples should be excluded because the preparation failed to tag cells rather than because there were no cells to tag. This idea behind the ZIP example is to show that the Bayesian method can allow for this sort of further investigation and, indeed, as the reviewer notes this sort of extended analysis is often bespoke, tailored to the data.

We have clearly failed to explain that the ‘standard workflow’ we propose replace the more traditional methods is the first one we describe, with the horseshoe prior; this produces better results on both datasets than the traditional approach. However, we also feel it is useful to show how a more tailored follow-on can be useful; we need to make it clear that this is intended as an illustration of an ‘optional extra’ rather than a part of the more straightforward ‘standard workflow’.

To make this clearer we have made altered the text in several locations:

• end of Introduction: added clarifying sentence “Here, our aim is to introduce a ‘standard’ Bayesian model for cell count data. We illustrate the application of this model to two datasets, one related to neural activation and the other to developmental lineage. For the second dataset, we also demonstrate a second example extension Bayesian model.”

• Section Hierarchical modeling: “Our goal in both cases is to quantify group differences in the data. We present a ‘standard’ hierarchical model. This model reflects the experimental features common to cell count experiments and reflects the hierarchical structure of cell count data; the standard model is designed to deal robustly and efficiently with noise. On some occasions, to reflect a specific hypotheses, the structure of a particular experiment or an observed source of noise, this model can be further refined or changed to target the analysis. We will give an example of this for our second dataset.”

• Section Horseshoe prior: “The alternative is via a flexible prior such as the horseshoe Carvalho et al., 2010; Piironen and Vehtari, 2017. This more generic option may be suitable as a default ‘standard’ approach in the typical case where outliers are poorly understood.”

• Discussion: word ‘standard’ added to sentence: “Our standard workflow uses a horseshoe prior, along with the partial pooling, this allows our model to deal effectively with outliers.”

• Discussion: modified sentence “The horseshoe prior model workflow we have exhibited here is intended as a standard approach.”

Indeed, because the horseshoe prior deals robustly with outliers, whereas the ZIP is intended to model the outliers, any substantial difference between the two should be examined carefully. The referee is right to point out that we have not explained this in any detail and has helpfully listed a few brain regions were there are differences. This is useful, particularly since the examples listed illustrate in a useful way the opportunities and hazards this sort of data presents. To address this, we have added a new version of Figure 6 to the revised manuscript

Previously Figure 6 showed two example brain regions: MPN and TMd. We have now added MH and SCH to the figure, and new text commenting on the insights the plots provide, both in the Results and Discussion.

**Reviewer #2 (Public review):**
“A clearer link between the experimental data and model-structure terminology would be a benefit to the non-expert reader.”

This is a very good point and we are acutely aware through our own work how difficult it can be moving between fields with different research goals, different scientific cultures and different technical vocabularies. Just as it can be difficult translating from one language to another without losing nuance and meaning, it can be a real challenge finding technical terms that are useful for the non-expert reader while retaining the precision the application requires! In the long run, we hope that, just as some of the very specialized vocabulary that surrounds frequentist statistics has become familiar to to the working experimental scientists, the precise terminology involved in Bayesian modelling will become familiar and transparent. However, in advance of that day, we have included a glossary of terms at the end of the main text, and have made numerous small tweaks to make sure that link between data and model terminology is clearer and better explained.

**Reviewer #1 (Recommendations fro the authors):**
(1) “I would strongly recommend that the authors include more case studies in the manuscript, and address the qualitative differences between the different versions of the model.”

We agree that our method will only become established when it is applied to more datasets, we hope to contribute to further analysis and we know other people are already using the approach on their own data. We do, however, feel that adding more datasets to this paper will make it longer and more complex; the plan, instead, is to use the method on novel datasets to test specific hypotheses, so that the results will include novel scientific findings as well as adding another illustration of the Bayesian approach applied to data that is already well studied.

(2) “Figure 6 is not discussed in the main text.”

We had discussed the results presented in Figure 6 in the second paragraph of the section “Case study two – Ontogeny of inhibitory interneurons of the mouse thalamus”, however the reviewer is right in that we did not directly refer to the Figure – this was an oversight. In any case, in the revised manuscript we present a new version of Figure 6 (in response to above comment), which is now explicitly cited in the text.

Revised Figure 6: Example data and inferences highlighting model discrepancies. On the left under ‘data’: boxplots with medians and interquartile ranges for the raw data for four example brain regions. The shape of each point pairs left and right hemisphere readings in each of the five animals. On the right under ‘inference’: HDIs and confidence intervals are plotted. Purple is the Bayesian horseshoe model, pink is the Bayesian ZIP model, and orange is the sample mean. The Bayesian estimates are not strongly influenced by the zero-valued observations (MPN, SCH, TMd) or large-valued outliers (MH) and have means close to the data median. This explains the advantage of the Bayesian results over the confidence interval.

**Reviewer #2 (Recommendations from the authors):**
(1) “This is a generally well-written methodology paper that also provides the underlying code as a resource. As a reviewer outside both cell-count modelling and hierarchical-Bayesian approaches (though with a general interest in the topics) I found the method a little difficult to follow and would have liked to have been left with a better understanding of how the method is applied to the data. For example, in Figure 1 we are introduced to brain region count, animal count, and “items”. Then in the next line: pooling, model, structure, population and etc in subsequent lines. It is not clear what the subscripts (the pools?) are referring to: are they different regions R or animals N? These terms need to be better linked to the data and/or trimmed. Having said that, the later results look like a solid contribution to the field with a significant reduction in uncertainty from the Bayesian approach over the frequentist one. A future version of the manuscript, therefore, would benefit from greater precision of language as well as an economy and greater focus of terms linking the method to the biology. This is particularly the case around the exposition parts in Figure 1, Figure 2, and the “Hierarchical modelling” section.”

This is another important point. We have now made numerous small changes to tighten up the text in the paper, in response to both this point and the next point.

(2) “Language throughout could be sharpened. Subjectivity like “surprising outliers” could be removed and quirky grammar like “often small, ten is a typical” improved. There are also typos “an rate” etc that should be tidied up.”

As per previous response, we have made numerous tweaks and small improvements and feel that the paper is stronger in this respect.

(3) “Figure 1 caption. “It is a spectrum that depends” Is spectrum the right word here? Also, “thicker stroke” what does this refer to? Wasn’t immediately clear. In A, why is the whole animal within the R bracket that signifies brain regions, and then the brain regions are within the N bracket that signifies whole animals? Apart from the teal colouring, what are the other coloured regions in the image referring to? Improving this first figure would greatly help a reader unfamiliar with the context of the approach.”

We have replaced the word “spectrum” with “continuum”. We have replaced “ Observed quantities have been highlighted with a thicker stroke in the graphical model.” with “The observed data quantities, y_i_ to y_n_, are highlighted with a thick line in the model diagrams”. We have added the following text to describe the red and green lines in panel A: “green and red lines indicate regions labeled as damaged”.

(4) “On P2 there is no discussion of priors when running through the advantage of the Bayesian approach. Is this a choice or an oversight? Priors do have a role in the later analysis.”

A short additional paragraph has been added to the introduction outlining the advantage of having a prior, but also noting that the obligation to pick a prior can be intimidating and that suggesting priors is one of the contributions of our paper: “A Bayesian model also includes a set of probability distributions, referred to as the prior, which represent those beliefs it is reasonable to hold about the statistical model parameters before actually doing the experiment. The prior can be thought of as an advantage, it allows us to include in our analysis our understanding of the data based on previous experiments. The prior also makes explicit in a Bayesian model assumptions that are often implicit in other approaches. However, having to design priors is often considered a challenge and here we hope to make this more straightforward by suggesting priors that are suitable for this class of data.”

(5) “On P4 more explanation would help greatly. Formulas like 23*10*4 or 50*6+50*4 are presented without explanation. What are the various numbers being multiplied? Regions, animals? Again, a clearer link between biological data and model structure would be advantageous.”

We have now modified this line to clearly state the numbers’ sources: “The index i runs over the full set of samples, which in this case comprises 23 brain regions ×10 animals ×4 groups ≈920 datapoints in the first study, and 50 brain regions × 6 HET animals + 50 brain regions × 4 KO animals ≈500 datapoints in the second.”

(6) “P6 and Results. Is it possible to show examples of the data set sampled from? Perhaps an image or two for the two experiments. Both Figures 4 and 5 as they currently are could be made slightly smaller to provide space for a small explanatory sub-panel. This would help ground the results.”

This is a good idea. We have now added heatmap visualisations of both entire datasets to revised versions of Figures 4 and 5 (assuming that this is what the reviewer was suggesting).